# Genetic Improvement and Nutrigenomic Management of Ruminants to Achieve Enteric Methane Mitigation: A Review

Vasfiye Kader Esen [1], Valiollah Palangi [2,*] and Selim Esen [3]

1  Sheep Breeding Research Institute, Balikesir 10200, Turkey
2  Department of Animal Science, Agricultural Faculty, Ataturk University, Erzurum 25240, Turkey
3  Balikesir Directorate of Provincial Agriculture and Forestry, Republic of Turkey Ministry of Agriculture and Forestry, Balikesir 10470, Turkey
*  Correspondence: valiollah.palangi12@ogr.atauni.edu.tr

**Abstract:** A significant portion of global greenhouse gas emissions is attributed to methane ($CH_4$), the primary greenhouse gas released by dairy animals. Thus, livestock farming has a new challenge in reducing enteric $CH_4$ for sustainability. In anaerobic microbial ecosystems such as the rumen, carbohydrates are converted into short-chain, volatile fatty acids that animals use for energy and protein synthesis. It is, therefore, essential to understand rumen physiology, population dynamics, and diversity to target methanogens. Thus far, numerous $CH_4$ mitigation strategies have been studied, including feeding management, nutrition, rumen modification, genetics, and other approaches for increasing animal production. As new molecular techniques are developed, scientists have more opportunities to select animals with higher genetic merit through next-generation sequencing. The amount of $CH_4$ produced per unit of milk or meat can be permanently and cumulatively reduced through genetic selection. Developing eco-friendly and practical nutrigenomic approaches to mitigating $CH_4$ and increasing ruminant productivity is possible using next-generation sequencing techniques. Therefore, this review summarizes current genetic and nutrigenomic approaches to reducing enteric $CH_4$ production without posing any danger to animals or the environment.

**Keywords:** methane; mitigation; genetic; nutrigenomic

## 1. Introduction

In response to the increasing population rate and growing animal protein demand, the ruminant husbandry or livestock sector is experiencing rapid structural and functional changes. In particular, consumption of livestock products has increased twofold in the past few decades, while milk supply and demand have increased by 26% and 2.4% annually, respectively, and are expected to increase by 25% in the near future [1,2]. In addition to supporting approximately 1.3 billion producers and retailers, the livestock sector contributes between 40 and 50% of the agricultural gross domestic product [2]. It is necessary to underscore, however, that livestock, although of great social and economic importance, is also one of the largest sources of greenhouse gases (GHG) worldwide, particularly enteric methane ($CH_4$) [3]. Approximately 13% of the world's GHG emissions come from livestock farming; $CH_4$ emissions, in particular, are important contributors since their global warming potential is 28 times higher than carbon dioxide's [4]. As this carbon cannot be utilized as an energy source in ruminants, $CH_4$ production has a negative impact on ruminant efficiency and profitability, as well as on the environment [5].

Ruminants produce $CH_4$ as a natural reaction during the microbial fermentation of its feed in the rumen [6]. In cattle, for example, enteric $CH_4$ is mostly produced in the rumen (87–90%) and in the large intestine (10–13%) [7,8]. Various microbial species participate in the conversion of feed material to $CH_4$ in the rumen; methanogenic archaea perform the final step [9]. There has been significant evidence that genetic variation of the host animals impacts bacterial community composition and $CH_4$ production [10]. Furthermore,

numerous studies have demonstrated that methane production is closely related to the weight of the animal, the amount of dry matter ingested, and the amount of gross energy ingested [11,12]. It is recognized that methane production is a hereditable trait, and that genetic selection for low-emitting ruminants is an effective mitigation option, assuming feed intake and animal productivity remain unchanged [13,14].

There is little information on how fermentative organisms interact with methanogens, but it is likely that other functional groups of microbes influence $CH_4$ production in rumens heavily. In addition to producing substrates necessary for the survival of methanogens, they also affect their numbers or other microbiota members responsible for producing methanogenic substrates [15–17]. To achieve a balance between food production and greenhouse gas emissions, microbiologists and nutritional scientists are investigating the genomes of the rumen microbiome to understand their function in terms of feed conversion efficiency, and plant cell wall degradation. Various strategies have been developed to mitigate enteric $CH_4$ emissions, such as alternate electron receptors [18], ionophore [19], enzymes [20], probiotic cultures [21], plant secondary metabolites [22]. Providing ruminants with forages that are more digestible or increasing their consumption of energy-dense feed is another effective $CH_4$ strategy [3]. Mitigating enteric $CH_4$ can also be achieved by improving pasture quality [1] and managing grazing [23], using antibiotics [24], vaccinations [25], or defaunation [26]. In any cases, it should be noted that developing strategies to reduce ruminant-derived $CH_4$ and alter microbiomes associated with ruminants will be one of the most significant challenges of the century and deserve a great deal of attention.

Despite the abundance of genetic improvement and nutrigenomic studies on $CH_4$ reduction, articles summarizing these studies are scarce. Hence, given the importance of the topic and the fast pace of growing knowledge in the area, in this article, the authors have tried to focus on genetic improvement and nutrigenomic management-based approaches to reduce enteric $CH_4$ emissions. In the current study, genetic strategies that can affect $CH_4$ production were classified into two major sections: 1–Manipulating ruminants via genetic selection and 2–Manipulation of the rumen microbiome.

## 2. Manipulating Ruminants via Genetic Selection

One of the methods, which can be classified in the manipulating the animal section, is genetically selecting ruminants to produce less $CH_4$. Genetic selection is an attractive solution, due to its cumulative and permanent nature; however, since selection is carried out over generations, it requires additive genetic variation and time to take effect [27]. Furthermore, genetic selection involves recording the $CH_4$ of a large number of animals, which is costly; therefore, to achieve a more accurate genetic evaluation, it is essential to use phenotypes that are precise and consistent [28].

In ruminants, approximately 98% of the $CH_4$ released is emitted directly from the rumen, absorbed from the rumen and hindgut into the blood, and exhaled from the lungs, while the remaining 2% is expelled as flatulence [27] and the rate of $CH_4$ emission varies with day [29], physiological state (growing, lactating and non-lactating) [30], and during lactation (early, mid, peak and late) [31], as well as between lactation periods [27]. In order to understand the implications of selection based on $CH_4$ emissions recorded at a particular point in time and to optimize selection strategies, it is essential to determine the relationship between $CH_4$ emissions recorded at various points during an animal's life and its phenotypic and genetic characteristics [27].

Identifying genetic correlations between one trait and other traits plays an important role in evaluating the effectiveness of the current breeding strategy. In order to reduce $CH_4$ emissions, we might select high-yielding animals for two main reasons. The first reason is that they are more productive, convert feed more efficiently, and require less maintenance. The second reason is that fewer animals with better productivity will require fewer animals to achieve a target production level [1]. The advantage of selecting high-yielding animals is that it allows cumulative and permanent changes; however, it depends on additive genetic variation and time to be effective since selection takes generations [27].

A repeatable and highly correlated method should be used with the existing traits in the selection index. Many researchers have speculated that a novel method of measuring the concentrations of $CH_4$ released by dairy cows during milking may provide valuable information regarding the daily emissions of individual dairy cows [32,33]. Moreover, the measurement of enteric $CH_4$ emissions in dairy cows is more straightforward than in cattle from another production system [33]. Typically, dairy cows are milked 3 times daily by standard milking parlors and 6 times daily by automatic milking systems [32]. Consequently, individual enteric $CH_4$ emissions can be monitored on an ongoing basis without the need for invasive techniques [32]. Thus, a number of researchers recommend the integration of measuring systems with automatic milking systems to be able to provide accurate, repeatable information on the amount of $CH_4$ emitted by dairy cows [32,34]. A number of techniques and devices have been used for this purpose, including respiration chambers [35], portable accumulation chambers [36], hexafluoride tracer methods [23], laser $CH_4$ detectors [37], micrometeorological methods [38], and more recently, measurements of milk in the mid-infrared range [39]. A detailed representation of the methane production and methane yield results obtained using different methods is given in Table 1.

**Table 1.** Comparison of techniques used to measure methane emissions from ruminants.

| Methods | Animal | Breed | Feed Intake | MeP | MeY | References |
|---|---|---|---|---|---|---|
| RC | Heifers | Hereford × Friesian | *ad libitum* (10.9–12.2 kg DM per day) | 265 | 24.5 | [40] |
| | Dairy cows | German Holstein (early lactation) | *ad libitum* TMR (grass silage, corn silage, barley straw, hay, concentrate, corn meal, canola seed meal, soybean meal, wheat, soybean oil) | 346.4 | 22.2 | [31] |
| | Beef steers | Brahman (*Bos indicus*) and Belmont Red (*Bos taurus x African Sanga*) | Rhodes grass pasture (*Chloris gayana*) grazed | 114.3 | 30.1 | [38] |
| | Lambs | Coopworth, Romney, Perendale, Texel, and composite breeds | *ad libitum* pasture allowance | 24 | 16 | [36] |
| PAC | Lambs | Coopworth, Romney, Perendale, Texel, and composite breeds | *ad libitum* pasture allowance | 7.5 | - | [36] |
| SF$_6$ | Heifers | Hereford × Friesian | *ad libitum* (10.9–12.2 kg DM per day) | 272 | 22.4 | [40] |
| | Cows | Angus | fed with 88% DM, 14% CP, 67% DM digestibility, and ME content of 9 MJ/kg DM | 132.6 | 21.9 | [41] |
| | Cows | Australian Holstein | fed a diet based on alfalfa supplemented with around 6 kg of crushed wheat per day | 110.5 | 17.5 | [41] |
| MHC | Heifers | Hereford × Friesian | *ad libitum* (10.9–12.2 kg DM per day) | 323 | 26.8 | [40] |
| | Dairy cows | German Holstein (early lactation) | *ad libitum* TMR (grass silage, corn silage, barley straw, hay, concentrate, corn meal, canola seed meal, soybean meal, wheat, soybeen oil) | 338.2 | 20.1 | [31] |
| MM | Beef steers | Brahman (*Bos indicus*) and Belmont Red (*Bos taurus x African Sanga*) | Rhodes grass pasture (*Chloris gayana*) grazed | 136.1 | 29.7 | [38] |
| MIR | Dairy cows | Spanish Holstein | - | 182.5 | 38.0 | [42] |

MeP: Methane production (g/d); MeY: methane yield (g/kg DMI (dry matter intake)); RC: respiration chambers; PAC: portable accumulation chambers; SF$_6$: hexafluoride tracer methods; MHC: mobile head-chamber (also known as GreenFeed); MM: micrometeorological methods; MIR: mid-infrared range method; TMR: total mixed ration; DM: dry matter; CP: crude protein; ME: metabolizable energy.

An accurate and preferably inexpensive phenotype is essential for making genetic evaluations relevant to the traits of interest. To genetically reduce GHG emissions, methane production (MeP, g/d), methane intensity (MI, g/kg at weight at test day (WT, kg)), methane yield (MeY, g/kg dry matter intake (DMI, kg/d)), and residual methane production (RMeP, g/d) are the most preferred phenotypes [28]. However, it might be challenging to properly incorporate MI, MeY, and RMeP into the selection index and explain to breeders why these phenotypes are the least frequently used. Therefore, the most correct way might be to use MeP in conjunction with its correlation structure with the milk yield, milk contents, live weight, and feed intake in the selection index [28]. Meanwhile, it should be noted that various factors may affect the production of $CH_4$ in individual cows, including their feed intake, dry matter content, feed composition, as well as fermentation rate in the rumen [43,44].

In contrast to MeP, several researchers have pointed out that RMeP is adjusted for traits that influence $CH_4$ outputs [28,45]. RMeP is computed as the difference between actual and predicted $CH_4$ outputs based on a subset of measured phenotypes [41]. As a statistical tool, RMeP offers advantages because its relationship with phenotypes used in its calculation is generally uncorrelated and it accurately predicts selection response [45]. Furthermore, Richardson et al. [45] reported that genetic interactions may still exist even after correcting RMeP traits for influential traits. A significant negative genetic correlation, for example, was found between the dry matter intake (DMI)-MeY (−0.60) and the energy-corrected milk (ECM)-MI (−0.73). A detailed comparison of genetic and phenotypic correlations between methane emission traits in ruminants is also provided in Table 2.

**Table 2.** Comparision of genetic and phenotypic correlations between methane emission traits in ruminants.

| Methods | Animal | Breed | GC | PC | References |
|---------|--------|-------|-----|-----|------------|
| RC | Lambs | Coopworth, Romney, Perendale, Texel, and composite breeds | BW-MeP: 0.83 BW-MeY: 0.02 | BW-MeP: 0.61 BW-MeY: 0.003 | [36] |
| RC | Cattle | Angus | - | MeP-DMI: 0.65, MeY-DMI: −0.02 | [35] |
| RC | Dairy cows | Holstein and Jersey [1] | MeP-DMI: 0.70, BW-MeP:0.54, ECM-MeP:0.66 | - | [46] |
| RC | Dairy cows | Holstein and Jersey [2] | MeP-DMI: 0.49, BW-MeP:0.24, ECM-MeP:0.52 | - | [46] |
| PAC | Lambs | Coopworth, Romney, Perendale, Texel, and composite breeds | BW-MeP: 0.59 | BW-MeP: 0.31 | [36] |
| MIR | Dairy cows | Spanish Holstein | MeP-RT: −0.43, MeP-MY: 0.21, MeP-PY: 0.31, MeP-FY: 0.29 | MeP-MY: 0.05, MeP-PY: −0.03, MeP-FY: 0.00 | [4,42] |
| SF$_6$ | Cows | Holstein and Angus | MeP: DMI: 0.83, MeP-BW: 0.80, MeY-DMI: 0.08, MeY-BW: 0.05 | MeP: DMI: 0.70, MeP-BW: 0.67, MeY-DMI: −0.00, MeY-BW: 0.04 | [41] |
| SF$_6$ | Dairy cows | - | MeP-DMI: 0.42, MEY-DMI: −0.60 | MeP-DMI: 0.49, MEY-DMI: −0.27 | [45] |
| Calculation | Heifers | Holstein-Friesian | - | MeP-FPCM: 0.26, MeP-DMI:0.99, MeP-RFI: 0.72 | [9] |

[1]: based on individual-level correlations; [2]: based on phenotypic level correlations; GC: genotypic correlation; PC: Phenotypic correlation; RC: respiration chambers; PAC: portable accumulation chambers; SF$_6$: hexafluoride tracer methods; MIR: mid-infrared range method; MeP: Methane production (g/d); MeY: methane yield (g/kg DMI (dry matter intake)); RT: rumination time (min/d); MY: milk yield (kg/d); PY: protein yield (kg/d); FY: fat yield (kg/d); FPCM: fat- and protein-corrected milk production (kg/d); RFI: residual feed intake (MJ/d).

A decrease in MeY has been observed with increasing DMI, even with small increases [36], while an increase in $CH_4/CO_2$ ratio was observed with increasing DMI [40]. Furthermore, Brask et al. [47] found that lactating dairy cows' $CH_4/CO_2$ ratio is higher at night than during the day, while it decreases with increasing time off pasture in sheep [48]. According to Robinson et al. [48], sheep's $CH_4$ emissions declined dramatically with increasing time away from pasture, while their $CO_2$ emissions declined only slightly.

Based on methane production rates, Kitelman et al. [49] identified three distinct ruminotypes in sheep: ruminotype Q (low-$CH_4$ production; higher abundances of propionate-producing e.g., *Quinella ovalis*), ruminotype S (low-$CH_4$ production; higher abundances of lactate- and succinate-producing bacteria, e.g., *Fibrobacter* spp., *Kandleria vitulina*, *Olsenella* spp., *Prevotella bryantii*, and *Sharpea azabuensis*), or ruminotype H (high-$CH_4$ production; higher abundances of *Ruminococcaceae*, *Lachnospiraceae*, *Catabacteriaceae*, *Coprococcus*, *Clostridiales*, *Prevotella*, *Bacteroidales*, and *Alphaproteobacteria*). Previous research also indicated that sheep selected for lower enteric $CH_4$ emissions have smaller rumens [50], a faster outflow of rumen liquor, and more VFA absorption gene expression in the rumen wall [51], which could lead to a reduction of ruminal VFA concentration. The muscle genes were identified as the best candidates as causal genes affecting $CH_4$ yield by [52]. In their study of the correlation between genetic parameters and $CH_4$ production, Jonker et al. [36] found that selectability for reduced $CH_4$ production can also lead to the selection of animals with a lower $CH_4$ yield due to dry matter intake [53] or animal genetics. It was found that $CH_4/DMI$ did not correlate clearly with fecal matter output [54]. According to Renand et al. [54], fecal matter output and retention have a weak correlation, while fecal output has a moderate correlation with $CH_4/DMI$. Moreover, $CH_4$ production is estimated to have a low genetic correlation with remaining traits [5].

Using genetic selection as a tool can reduce $CH_4$ emissions and improve ruminant energy efficiency simultaneously without negatively affecting their important economic traits [45]. Breeding objectives for ruminants should include reducing $CH_4$ emissions while preserving economic traits that are important to sustainability.

## 3. Manipulation of the Rumen Microbiome

### 3.1. Rumen Microbiome

Recently, efforts have been directed toward characterizing the rumen microbiome and its function in order to implement nutritional and selective breeding strategies to alter it. Both the host and the ruminal microbiota affect livestock traits such as efficiency and sustainability, including $CH_4$ production, and are partly controlled by the host genotype [55]. The rumen microbiota, however, is highly dependent on the ruminant species, diet, and geographic location, leading to different rumen microbiome profiles and dietary nutrient utilization in ruminants from tropical and temperate environments [56]. Although these factors are present, microbial communities are generally stable due to ecological redundancy and resilience to external and internal perturbations [7].

In newborn ruminants, gastrointestinal tracts (GITs) are considered sterile upon birth, but methanogens and fibrolytic bacteria appear within 20 min of birth [57]. A day after parturition, cellulolytic bacteria, such as *Ruminococcus flavefaciens*, *Ruminococcus albus*, *Prevotella* species are predominant in the rumen microbiota [58], whereas xylanase and amylase (carbohydrate degrading enzymes) activity, as well as VFAs production, are evident within 2 days [59]. Furthermore, anaerobic fungi and methanogens begin to colonize the neonatal rumen around 8 to 10 days postpartum, but protozoa do not appear until 15 days [24]. Among the microbiomes found in adult's GITs of ruminants, anaerobic bacteria are the most abundant ($10^{10-11}$/mL), followed by archaea (methanogens; $10^{8-9}$/mL), protozoa ($10^6$/mL), and fungi ($10^6$/mL) that digest feed together [60]. Compared to mature animals, pre-ruminant methanogenic archaea produce more $CH_4$ from a wide range of substrates, possibly because the *Methanobacteriaceae* dominate the rumen around 3 months of age [61]. In later stages of life, changing the rumen's microbial ecology is more difficult [62]. Thus, several studies have shown that a favorable mi-

crobiome can be implanted into the rumen microbiome early in life through dietary or management intervention [24,63].

Several studies have indicated that the seven most abundant bacterial groups (*Prevotella*, *Butyrivibrio*, and *Ruminococcus*, *Lachnospiraceae (unclassified)*, *Ruminococcaceae*, *Bacteroidales,* and *Clostridiales*) are found in many species of ruminants, accounting for 67.1% of all sequenced data, and are defined as "**core bacterial microbiomes**" at the genus or higher level [64,65]. Except for *Butyrivibrio*, none of these groups are adequately represented by characterized cultures, nor are their functions understood [66]. The largest clade of archaea, *Methanobrevibacter gottschalkii* and *Methanobrevibacter ruminantium*, represents 74% of all archaea. On the other hand, an archaeal community in the rumen was composed of five dominant methanogen groups, as well as one *Methanosphaera* spp. and two *Methanomassiliicoccaceae*-associated groups, accounting for 89.2%. This implies that rumen archaea have less diversity than rumen bacteria [65,67].

### 3.2. Manipulation of the Rumen Microbiome via Nutrigenomic Approaches

Despite the lack of clarity regarding the relationship between fermentative organisms and methanogens, there are a number of functional groups of microbes that may have a significant impact on $CH_4$ production in the rumen, either by producing substrates that promote methanogen survival or by altering the number of methanogens or other microbes that produce methanogenic substrates [15]. Therefore, changing the dynamics of rumen fermentation by modifying other microbial groups may be an effective method for reducing $CH_4$ production.

Rumen hydrogen production is directly influenced by the pattern of fermentation of volatile fatty acids (VFAs), in other words ruminal VFA profile is an indicator of microbial activity [68]. As a result of glycolysis and final synthesis of VFA in the rumen, hydrogen is produced. For instance, 1 mol of acetate yields 2 mol of hydrogen, whereas 1 mol of propionate yields only 1 mol of hydrogen [69]. It is possible, therefore, to reduce $CH_4$ emissions via interventions affecting the rumen microbiome's acetate/propionate ratio [70], increasing bacteria species that compete for hydrogen (such as sulfate reducers and acetogens) [71], or inhibiting protozoa, which can cause hydrogen production to be reduced [72].

As an ionophore (which have antibiotic capacity), monensin is used to prevent ketosis in dairy cows and to increase production [73]. It alters the rumen microbiota, leading to increased hepatic gluconeogenesis and, therefore, an increase in the animal's energy supply when added to the diet, which results in an increase in propionate production [74]. In order to determine the differential effects of monensin and a mixture of essential oils on rumen microbiota composition in transition dairy cows, an experiment was conducted. Monensin decreased the relative abundance of 23 OTUs from the phyla *Bacteroidetes* and *Firmicutes*, whereas it increased the abundance of 10 OTUs from the phyla *Actinobacteria*, *Bacteroidetes*, *Cyanobacteria*, and *Firmicutes* [72]. As a result of monensin supplementation, the *Butyrivibrio* genus (which produces butyrate and acetate) was inhibited and propionate production was increased, whereas *Prevotella* and *Ruminococcaceae* (succinate and propionate producers, respectively) increased in abundance due to the rapid conversion of succinate to propionate by succinate-decarboxylating bacteria in the rumen [72]. It has previously been demonstrated that *Prevotella* species compete with methanogens for hydrogen use, rerouting it to propionic acid production and reducing methanogen availability [75].

A reserve polysaccharide in plants, inulin influences the proliferation of probiotics, such as *Lactobacillus* and *Bifidobacteria*, and inhibits bacterial pathogens, such as *Clostridium pneumonia*, *Enterococcus* and mold [76]. The effects of inulin on goat rumen fermentation and microbial growth were investigated by Zhao et al. [77] using rumen simulation technology. It was determined that inulin treatment decreased acetate concentrations, acetate ratios, and $CH_4$ production, while it increased butyrate concentrations. Furthermore, inulin inhibited *Fibrobacter succinogenes* and *Ruminococcus flavefaciens* growth, indicating that it might

suppress the growth of rumen bacteria that decompose cellulose. In a study of finishing Simmental × Luxi crossbred beef steers fed a high or low-concentrate diet supplemented with 2% inulin (*wt/wt*), Tian et al. [78] demonstrated similar results regarding rumen fermentation and bacterial microbiota. The addition of inulin to a finishing beef diet resulted in a shift in the fermentation of acetate to propionate and butyrate, resulting in an increase in the α-diversity indexes of rumen bacteria, particularly *Bacteroides* and *Firmicutes*, which were significantly more abundant [78]. Furthermore previous reports has also demonstrated that cellulolytic *Ruminococcus* species (*Ruminococcus flavefaciens* and *Ruminococcus albus*)—along with *Butyrivibrio fibrisolvens*—were unable to grow in medium containing long-chain polyunsaturated fatty acids (such as docosahexaenoic acid or eicosapentaenoic acid) [79,80]. Additionally, Burdick et al. [81] found that *Methanobrevibacter gottschalkii* relative abundance increased while *Methanosphaera* spp. ISO3-F5 relative abundance decreased, but the ratio *Methanobrevibacter gottschalkii*: *Methanobrevibacter ruminantium*, associated with lower $CH_4$ emissions [82], did not change in lactating Holstein cows supplemented with medium-chain fatty acids.

During the course of researching 12 Nordic macroalgae species for anti-methanogenic properties, Pandey et al. [83] found that polyphenol-rich brown species, such as *Fucus vesiculosus* and *Ascophyllum nodosum*, significantly reduced feed degradability due to suppressed cellulolytic bacteria (*Ruminococcus* spp., *Lacnospiraceae* spp., *Rikenellaceae* RC9). These two macroalgae have been shown to reduce $CH_4$ production by 62.6 and 48.2%, respectively, and to reduce methanogenic archaea in rumens (such as *Methanobrevibacter* spp.). On the other hand, the reduction was not directly correlated with polyphenol concentrations overall.

Aside from converting carbohydrates into succinate and acetate, the *Succinivibrionaceae* family (*Succinivibrio*, *Ruminobacter*, *Anaerobiospirillum*, and *Succinimonas*) also produces hydrogen and acetate, which reduces $CH_4$ emissions [84]. Additionally, previous research suggests that feed efficiency of beef cattle [85] and milk protein of dairy cows [86] are highly related to the *Succinivibrio* spp. population in the rumen due to its function of producing succinate, the precursor of propionate.

Dairy cow diets are often supplemented with lipids to increase energy content; however, such lipid sources can change the composition of rumen bacteria and may affect biohydrogenation processes. For instance, Vargas-Bello-Perez et al. [87] investigated the effects of two dietary lipids on bacterial populations and fatty acid profiles in non-lactating Holstein cows' rumen digesta by utilizing soybean oil (an unsaturated oil source) and hydrogenated palm oil (a saturated oil source) and they found that dietary treatments had no effect on *Fibrobacter succinogenes*, *Butyrivibrio proteoclasticus*, and *Anaerovibrio lipolytica* loads. The same authors noted that with hydrogenated palm oil supplementation, the load of *Prevotella bryantii* significantly increased compared with control. In order to develop effective $CH_4$ mitigation strategies, Gruninger et al. [88] added 3-nitrooxypropanol (3-NOP), canola oil, and their combination to a high-forage diet (90% barley silage) of beef cattle. The authors noted that 3-NOP decreased the abundance of *Methanobrevibacter* and increased the abundance of *Bacteroidetes*, whereas canola oil significantly reduced the abundance of protozoa and fiber-degrading microbes in rumens, but did not significantly alter the abundance of rumen methanogens [88]. On the other hand, soybean or linseed oil (4% of DM) appeared to be more effective at decreasing *Butyrivibrio fibrisolvens*, *Ruminococcus albus,* and *Fibrobacter succinogenes* by 18, 42, and 67%, respectively [89]. An increase in the relative abundance of *Prevotella* and *Dialister* bacteria was also observed following the addition of oregano essential oil, which indicates that it has the potential to manipulate ruminal fermentation and reduce $CH_4$ emissions [90].

Research has also shown that fiber digestion and $CH_4$ production are related, particularly in the rumen where fibrolytic bacteria are involved in $H_2$ production (e.g., *Ruminococcus*, *Eubacterium*) and consumption (e.g., *Bacteroidetes*) [91]. Acetic acid production in the rumen is also closely linked with the production of $H_2$, which increases the substrate availability for $CH_4$ production [92]. A reduction in fiber content in diet is known to decrease *Bacteroides* abundance, as Pitta et al. [93] demonstrated. In addition, Mu et al. [94] reported that reducing fibrous substrate could be one explanation for *Fibrobacter* spp. (associated with cellulose degradation) and *Alistipes* spp. (associated with oligosaccharide degradation and butyric acid production [95]) declines in lactating Holstein cows fed high-grain diets. It is possible that *Fibrobacter succinogenes*'s cellulolytic activity stimulated *Prevotella ruminicola*'s production of propionic acid, since these bacteria consume only released sugars during cellulose digestion [95,96]. The rumen acetate proportion and the growth of *Ruminococcus albus* increased when forage concentrations were high in mixed diets [97]. Calves supplemented with *Prevotella* and cellulolytic bacteria (*Ruminococcus flavefaciens* and *Ruminococcus albus*) also showed similar results [98]. Furthermore, *Prevotella* may dominate the rumen in animals fed high-fiber diets, whereas *Bacteroidetes* may occupy a greater proportion in hay-supplemented animals, leading to increased rumen size [99,100].

## 4. Future Perspectives

In line with ruminal methane production reduction efforts, numerous methane mitigation strategies have been investigated, reported, and suggested by scientists to the livestock industry. Yet, the majority of studies have solely reported the impact of proposed strategies on the final product (methane production); the process of methane reduction has often been ambiguous. Thus, clarifying the process of methane reduction in the rumen is essential. Despite the positive effects on reducing enteric $CH_4$ emissions, the genetic selection method leads to decreased rumen volume and increased passage rate. Therefore, it may result in decreased ruminal fermentation and production of VFA, and finally will reduce the efficiency of microbial fermentation. Manipulation of the rumen microbiome seems to be a better method, but it requires further studies to investigate its different aspects.

**Author Contributions:** Conceptualization, V.P.; methodology, V.K.E., V.P. and S.E.; writing—original draft preparation, V.K.E., V.P. and S.E.; writing—review and editing, V.K.E., V.P. and S.E.; visualization, S.E. All authors have read and agreed to the published version of the manuscript.

**Funding:** This research received no external funding.

**Institutional Review Board Statement:** Not applicable.

**Informed Consent Statement:** Not applicable.

**Data Availability Statement:** Not applicable.

**Conflicts of Interest:** The authors declare no conflict of interest.

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
