# Peer review of "Genetic Improvement and Nutrigenomic Management of Ruminants to Achieve Enteric Methane Mitigation: A Review"

_methane, doi:10.3390/methane1040025_

Round 1

Reviewer 1 Report

- You said, "One of the methods, which can be classified in manipulating the animal section, is genetically selecting ruminants to produce less CH4”. How do you do manipulation, and which genetic factors? Also, your paragraph description (LN: 68-87) is unclear. Please verify and fix them!

- Which ruminant and which method produces less CH4 for ruminants? How about other ruminants, such as beef cattle? Why are you not concern about them as well? Please add some reason!

- LN 88-98: Are you comparing methods to determine methane gas production between automatic milking systems (as you claimed the accurate methods) and others? Are mid-infrared range, respiratory chambers and others not accurate and relevant to use? Please add some reasons and fix them.

- Figure 1, You adapt and interpret the data from one article and claim that between methods have some differences compared to the automatic milking system is irrelevant. It would be best to compare with other studies so your argument can be accepted. This case is similar to figure 2 and Figure 3. 

- The flow of the story from paragraphs 104-114 needs to be clarified. I do not understand. Please fix them.   

- No explanation of unit abbreviations that you mentioned in the articles. Please explain the first time you mention them.

- LN 167-168: How?

- Concerning your statement on butyrivibrio, your statement on line 202 and 235 are contradictory. Please fix them. 

- Some of your statements about the relationship between bacteria on metabolism and methane production need to be revised. Please fix them. 

For a review, especially concerning comparing between numerical data, it is not only need one articles (mentioned in Figures)  just to strengthen the argument in the review articles. It is need comparison studies.     

Author Response

Response to reviewer 1

Thanks to the reviewer for thoroughly examining our manuscript and providing very useful feedback for our revisions. In response to the reviewer's suggestions and comments, the authors have modified the manuscript. In the revised manuscript, the modified words were highlighted in red. We are grateful for the constructive comments and suggestions offered by the reviewer. Below you will find our responses to your comments.

Major comments:

  1. You said, "One of the methods, which can be classified in manipulating the animal section, is genetically selecting ruminants to produce less CH4”. How do you do manipulation, and which genetic factors? Also, your paragraph description (LN: 68-87) is unclear. Please verify and fix them!

Reply to reviewer: Thank you very much for pointing this out. We have made the necessary corrections in accordance with your suggestions. To verify, please refer to L266-270 in the revised manuscript.

  1. Which ruminant and which method produces less CH4 for ruminants? How about other ruminants, such as beef cattle? Why are you not concern about them as well? Please add some reason!

Reply to reviewer: While the focus of his review has been on cattle (Ruminants that have already been genetically modified), there are other ruminants of importance too, such as buffalo, sheep, goats and camels (Of course, according to the available references, we tried to mention them as well). But first, due to the fact that beef cattle often receive concentrated feed, therefore, the energy cycle is more inclined towards propionic production, as a result of which hydrogen is out of the reach of methanogens, and therefore the share of methane production in them is less. Additional information is available in our previous article (https://doi.org/10.3390/su142013229).

  1. LN 88-98: Are you comparing methods to determine methane gas production between automatic milking systems (as you claimed the accurate methods) and others? Are mid-infrared range, respiratory chambers and others not accurate and relevant to use? Please add some reasons and fix them.

Reply to reviewer: Thank you very much for pointing this out. We have made the necessary corrections in accordance with your suggestions. To verify, please refer to L113-122 in the revised manuscript.

  1. Figure 1, You adapt and interpret the data from one article and claim that between methods have some differences compared to the automatic milking system is irrelevant. It would be best to compare with other studies so your argument can be accepted. This case is similar to figure 2 and Figure 3.

Reply to reviewer: Thank you very much for pointing this out. All figures have been removed and two different tables have been inserted in accordance with your suggestions. To verify, please refer to L97 and L158 in the revised manuscript.

  1. 5. The flow of the story from paragraphs 104-114 needs to be clarified. I do not understand. Please fix them.

Reply to reviewer: Thank you very much for pointing this out. The sentence related to Figure 2 has been removed. To verify, please refer to L129-133 in the revised manuscript.

  1. No explanation of unit abbreviations that you mentioned in the articles. Please explain the first time you mention them.

Reply to reviewer: Thank you very much for pointing this out. We have made the necessary corrections in accordance with your suggestions. To verify, please refer to L129-133 in the revised manuscript.

  1. LN 167-168: How?

Reply to reviewer: Thank you very much for pointing this out. We fix them. To verify, please refer to L185-186 in the revised manuscript.

  1. Concerning your statement on butyrivibrio, your statement on line 202 and 235 are contradictory. Please fix them.

Reply to reviewer: Thank you very much for pointing this out. We fix them. To verify, please refer to L219-220 and L251-256 in the revised manuscript.

  1. Some of your statements about the relationship between bacteria on metabolism and methane production need to be revised. Please fix them

Reply to reviewer: Thank you very much for pointing this out. We fix them.

  1. For a review, especially concerning comparing between numerical data, it is not only need one articles (mentioned in Figures) just to strengthen the argument in the review articles. It is need comparison studies.

Reply to reviewer: Thanks for your suggestion. All figures have been removed and two different tables have been inserted in accordance with your suggestions. To verify, please refer to L97 and L158 in the revised manuscript.

Reviewer 2 Report

Point 1: Line 242-247: Is the effect of inulin in in vivo experiments consistent with the results obtained in experiments where ruminal simulation was used?

Point 2: Line 274-278: In vitro techniques are of great importance to generate an overview of ruminal fermentation, however, with in vivo experiments, the results of ruminal fermentation can be very distant from those observed in the laboratory. Therefore, it is suggested to review current literature on the use of lipids in the diet and their response on ruminal fermentation with in vivo experiments. This will allow greater clarity in the text.

Author Response

Response to reviewer 2

Thanks to the reviewer for thoroughly examining our manuscript and providing very useful feedback for our revisions. In response to the reviewer's suggestions and comments, the authors have modified the manuscript. In the revised manuscript, the modified words were highlighted in red. We are grateful for the constructive comments and suggestions offered by the reviewer. Below you will find our responses to your comments.

Major comments:

  1. Line 242-247: Is the effect of inulin in in vivo experiments consistent with the results obtained in experiments where ruminal simulation was used?

Reply to reviewer: Thank you very much for your questions. Yes, the inulin effects observed in the experiments involving rumen simulation are consistent with those observed in the experiments conducted in vivo. To verify, please refer to L266-270.

  1. Line 274-278: In vitro techniques are of great importance to generate an overview of ruminal fermentation, however, with in vivo experiments, the results of ruminal fermentation can be very distant from those observed in the laboratory. Therefore, it is suggested to review current literature on the use of lipids in the diet and their response on ruminal fermentation with in vivo experiments. This will allow greater clarity in the text.

Reply to reviewer: Thank you very much for pointing this out. The results of the in vitro studies have been excluded and the in vivo studies have been placed according to your suggestions. To verify, please refer to L294-308.

Reviewer 3 Report

The authors present a manuscript "Genetic Improvement and Nutrigenomic Management of Ruminants to Achieve Enteric Methane Mitigation: A review" on a very relevant and current topic. The paper is well written and correctly structured for what is likely to be published.

However, as this is a review, it would be important to incorporate in the introduction the role of different ruminant production systems in emissions (positive vs. negative effects). And the strategies based on homoacetogens, vaccination, bacteriophages and enzyme inhibitors should also be mentioned in the text, albeit briefly.

Author Response

Response to reviewer 3

Thanks to the reviewer for thoroughly examining our manuscript and providing very useful feedback for our revisions. In response to the reviewer's suggestions and comments, the authors have modified the manuscript. In the revised manuscript, the modified words were highlighted in red. We are grateful for the constructive comments and suggestions offered by the reviewer. Below you will find our responses to your comments.

Major comments:

  1. The authors present a manuscript "Genetic Improvement and Nutrigenomic Management of Ruminants to Achieve Enteric Methane Mitigation: A review" on a very relevant and current topic. The paper is well written and correctly structured for what is likely to be published.

However, as this is a review, it would be important to incorporate in the introduction the role of different ruminant production systems in emissions (positive vs. negative effects). And the strategies based on homoacetogens, vaccination, bacteriophages and enzyme inhibitors should also be mentioned in the text, albeit briefly.

Reply to reviewer: Thank you very much for pointing this out. In the introduction section, we address the role of different ruminant production systems (positive versus negative effects) and the various strategies available for mitigating enteric methane emissions. To verify, please refer to L30-39 and L62-68.

Round 2

Reviewer 1 Report

No more comment